# Exploring the Multifunctional Role of Alpha-Fetoprotein in Cancer Progression: Implications for Targeted Therapy in Hepatocellular Carcinoma and Beyond

**DOI:** 10.3390/ijms26104863

**Published:** 2025-05-19

**Authors:** Hyunjung Kim, Minji Jang, Eunmi Kim

**Affiliations:** Cancer Molecular Biology Branch, Division of Cancer Biology, National Cancer Center, Goyang-si 10408, Republic of Korea; jung@ncc.re.kr (H.K.); mji9997@ncc.re.kr (M.J.)

**Keywords:** alpha-fetoprotein, tumor progression, hepatocellular carcinoma, cancer type, target cancer therapy

## Abstract

Alpha-fetoprotein (AFP) is a well-known biomarker for liver cancer, and its clinical utility is widely recognized. Recent studies have revealed that AFP plays a multifaceted role in various malignant tumors, including liver cancer. This suggests that AFP is not merely a biomarker but also contributes significantly to the complex process of tumor formation, emphasizing the importance of targeting AFP in therapeutic approaches. Consequently, innovative research and development are essential to overcome the current limitations of AFP-targeted therapies, enhance treatment efficacy, and minimize side effects. This review explores the role of AFP in cancer development and progression, highlights the biological functions of AFP and related pathways, and discusses the clinical implications of AFP-targeted therapies. Ongoing research on AFP will significantly contribute to our understanding of the biological mechanisms of cancer and aid in developing effective and safe treatments. Ultimately, advancements in AFP-targeted therapeutic approaches are expected to play a crucial role in the future of cancer management.

## 1. Introduction

Alpha-fetoprotein (AFP) is structurally a protein consisting of a single polypeptide chain of 609 amino acids with a molecular weight of approximately 69 kDa, and it shares a structural similarity with albumin. This protein is divided into three structural domains: the N-terminal domain (Domain I, amino acids 1–210), the central domain (Domain II, amino acids 211–402), and the C-terminal domain (Domain III, amino acids 403–609). These three domains are connected by disulfide bonds, forming a V-shaped structure, and each domain carries out specific biological functions. In particular, the C-terminal domain contains a leucine zipper-like structure that allows it to bind to signaling proteins and receptors, thereby promoting tumor growth, mediating intracellular transport, and regulating cellular signaling functions [1,2]. Figure 1 provides a schematic overview of the structure of human AFP.

AFP is widely recognized as a significant biomarker for liver cancer, particularly hepatocellular carcinoma (HCC), and its utility has been demonstrated in various clinical settings [3,4,5,6]. Initially discovered in fetal serum, AFP plays a crucial role in ensuring placental function and promoting fetal growth. It is primarily produced in the fetal liver and yolk sac [7,8,9]. This protein is essential for various biological processes, including the regulation of cell proliferation and differentiation, underscoring its significance during the early developmental stages. In adults, AFP expression is typically decreased, reflecting the differentiation stability of mature tissues, and its serum level generally shows a half-life of approximately five days [1,10]. AFP is a glycoprotein that can be classified into three glycoforms (AFP-L1, AFP-L2, and AFP-L3) based on its binding affinity to lens culinaris agglutinin (LCA). Among these, AFP-L3, which shows strong binding to LCA, is known to be specifically elevated in HCC. In Japan, AFP-L3 has been proposed and is currently used in clinical practice for the diagnosis and prognosis of HCC [11,12,13].

However, recent studies have indicated that the role of AFP extends far beyond its traditional characterization as a liver cancer marker. Elevated AFP levels are increasingly being associated with tumor progression, angiogenesis, metabolic reprogramming, immune evasion, and drug resistance in several malignancies [1,14,15,16,17,18]. These findings demonstrate that AFP is not merely a passive biomarker but actively participates in tumorigenesis. Notably, these characteristics of AFP are not restricted to liver cancer but are also closely associated with poor prognosis, inadequate treatment response, and increased metastatic potential in various cancers, including germ cell tumors, colorectal cancer, and gastric cancer [19,20,21,22]. These findings highlight the importance of AFP in cancer treatment strategies and its potential as a therapeutic target.

Based on these insights, this review aims to comprehensively explore the role of AFP in the development and progression of cancer, highlight the biological functions of AFP and related pathways, and discuss the clinical implications of AFP-targeted therapies. This provides a foundation for the effective utilization of AFP in future treatments and explores new approaches for cancer therapy.

## 2. AFP as a Tumor Progression Factor in HCC

### 2.1. Tumor Growth and Proliferation

While serum AFP levels are known to be significant prognostic factors in patients with HCC and have been the primary focus of research [15,23], recent studies have increasingly revealed that cytoplasmic AFP also plays an important role in regulating the growth signaling pathways of HCC [14,24,25]. Specifically, cytoplasmic AFP activates the PI3K/Akt/mTOR signaling pathway and reduces cell autophagy, thereby playing a crucial role in cell growth, survival, proliferation, and tumor progression [26]. In addition, AFP interacts with retinoic acid receptors to prevent them from entering the nucleus, thereby promoting the expression of Bcl-2, an anti-apoptotic protein [27]. To further maintain cancer growth and progression, AFP suppresses the human antigen R (HuR)-mediated apoptotic pathway involving Fas/FADD. It also upregulates KRAS, cyclic adenosine monophosphate, and protein kinase A and increases cytosolic Ca^2+^ levels [14,28]. Furthermore, AFP is known to shield HCC cells from TNF-induced cell death and promote escape of tumor cells from lymphocyte cytotoxic cells via the caspase enzyme pathways [29,30]. Notably, AFP physically binds to caspase-3, thereby blocking the apoptotic signaling cascade from caspase-8 to caspase-3 [31]. Overall, these findings highlight the potential of targeting AFP-related pathways as a therapeutic strategy against HCC.

### 2.2. Angiogenesis

Angiogenesis is essential for tumor growth and depends on the production of angiogenic factors, including vascular endothelial growth factor (VEGF) [32,33,34]. In human HCC samples, differential molecular patterns based on serum AFP levels were identified through whole-genome expression analysis, which revealed that AFP-high tumors exhibited significant activation of VEGF signaling. In particular, the overexpression of VEGFB and placental growth factor ligands observed in AFP-high tumors may result in enhanced activation of VEGFR1 and, at the same time, prevent VEGFA from binding to VEGFR1. The competition of VEGFA with other ligands could favor its binding to VEGFR2, ultimately leading to its subsequent activation and release of pro-angiogenic signals [35]. Furthermore, given the link between chronic inflammation and angiogenesis, AFP influences the production of inflammatory cytokines, such as interleukin-6 (IL-6), transforming growth factor beta 1 (TGF-β1), C-X-C chemokine receptor type 4 (CXCR4), and nuclear factor kappa-light-chain-enhancer of activated B cells, thereby contributing to this process [1]. Through these mechanisms, AFP plays a significant role in promoting angiogenesis in tumors, thereby supporting tumor growth and progression by ensuring an adequate supply of oxygen and nutrients.

### 2.3. Epithelial-Mesenchymal Transition (EMT) and Invasion

High serum concentrations of AFP positively correlate with metastasis in patients with HCC [36,37]. In such cases, AFP promotes the malignant behavior of HCC cells by inhibiting the activity of PTEN and activating the PI3K/Akt pathway, thereby inducing the expression of proteins essential for cell metastasis, such as keratin 19, matrix metalloproteinase 2/9, and CXCR4 [26,36]. The Wnt signaling pathway promotes the stabilization of β-catenin, a central component of the EMT pathway, which can also contribute to cancer metastasis. In the absence of Wnt ligands, β-catenin is typically targeted for degradation [38]. However, AFP can inhibit this degradation by facilitating the accumulation of β-catenin in the cytoplasm, which allows it to translocate to the nucleus [39]. The activation of Wnt/β-catenin signaling plays a crucial role in repressing epithelial markers, such as E-cadherin, while promoting the expression of mesenchymal markers, such as vimentin and zinc finger E-box binding homeobox 1. This transition from an epithelial to a mesenchymal phenotype is essential for cancer progression [40,41,42,43,44]. Additionally, AFP is associated with various microRNAs (miRNAs) that regulate EMT and influence key cellular processes, such as migration, invasion, and metastasis. For example, AFP can regulate the expression of miR-29, and serum levels of AFP in patients with HCC are significantly correlated with several miRNAs, including miR-451a, miR-483-5p, miR-423-3p, miR-221-3p, let-7b-5p, miR-301a-3p, miR-410, and miR-382-5p [45]. This highlights the close interaction between AFP and miRNAs during the metastatic process. Further studies are needed to fully elucidate the impact of their relationship on cancer metastasis.

### 2.4. Metabolic Reprogramming

#### 2.4.1. Influence on Glycolysis and Gluconeogenesis

Cancer cells exhibit metabolic differences from normal cells that allow them to produce energy and grow rapidly [46,47]. In many tumorigenic processes, AFP activates the PI3K/Akt/mTOR signaling pathway [26]. This signaling can enhance basic glycolytic activity in HCC by specifically improving the expression and activity of key glycolytic enzymes, such as hexokinase and phosphofructokinase [48,49,50]. Consequently, this promotes the conversion of glucose to pyruvate, which is crucial for energy production by cancer cells. Moreover, AFP has been shown to stimulate glucose transporters (GLUTs), particularly GLUT3, thereby increasing glucose uptake by cancer cells and facilitating glycolysis even under hypoxic conditions [51,52]. Several studies have also investigated the effects of AFP on glucose metabolism in immune cells of patients with HCC. When AFP derived from patients with HCC is taken up by dendritic cells (DCs), it induces a switch to glycolysis, resulting in increased glucose uptake and lactate secretion. This metabolic skewing contributes to immune suppression by increasing the expression of CD14 and programmed death-ligand 1 in DCs [52].

#### 2.4.2. Regulation of Lipid and Energy Metabolism

Studies investigating the effects of tumor-derived AFP (tAFP) on lipid metabolism in DCs have shown that it downregulates several key genes involved in fatty acid metabolism, including pyruvate dehydrogenase, ATP-citrate lyase, acetyl-CoA carboxylase, fatty acid synthase, lipoprotein lipase, CD36, and carnitine-acylcarnitine translocator. This suggests that tAFP negatively affects fatty acid uptake and oxidation. Furthermore, it inhibits mitochondrial oxidative phosphorylation by reducing the levels of mitochondrial protein cytochrome c oxidase [52]. Also, tAFP decreases the expression of the mitochondrial biogenesis transcription factor, peroxisome proliferator-activated receptor gamma coactivator 1 alpha, in patients with HCC [17]. Fatty acid synthesis is an essential cellular process for generating lipids that can be utilized for membrane synthesis or ATP production [53]. Therefore, the negative effects of tAFP-induced downregulation of fatty acid synthesis and inhibition of oxidative phosphorylation may contribute to the impaired development and differentiation of DCs, potentially leading to the development of HCC.

#### 2.4.3. Regulation of Amino Acid Metabolism

Although there is limited evidence of AFP directly influencing amino acid metabolism, AFP levels may reflect changes in amino acid metabolism in the liver. Interactions between branched-chain amino acid (BCAA) metabolism and AFP levels have been observed in patients with hepatitis C virus infection, suggesting that increased breakdown of BCAAs might provide an enhanced energy supply to rapidly dividing cancer cells. Both AFP and BCAAs participate in the metabolic reprogramming of hepatocytes, with BCAAs affecting serum AFP expression [54,55]. Further research focused on elucidating these relationships could provide deeper insights into the effects of AFP on amino acid metabolism within the liver.

### 2.5. Immune Evasion

#### 2.5.1. Effects on T Cells

A specific epitope within the AFP sequence is recognized by human CD4^+^ T cells [56]. This recognition is important as it suggests that AFP may be a target for T cell-mediated immune responses. Upon recognizing the AFP epitope, CD4^+^ T cells differentiate into TGF-β-producing cells [57]. TGF-β, a cytokine that promotes an immunosuppressive microenvironment, can facilitate tumor growth by enhancing the differentiation of regulatory T cells [58]. Therefore, the production of TGF-β by CD4^+^ T cells in response to AFP suggests a potential mechanism by which tumors expressing AFP can evade immune detection.

#### 2.5.2. Effects on Macrophages

Tumor-associated macrophages (TAMs) play crucial roles in promoting HCC invasion and metastasis, immune escape, matrix remodeling, EMT, lymphangiogenesis, angiogenesis, and drug resistance. Generally, TAMs exhibit characteristics of M2 macrophages [59,60]. Recent studies have shown that purified human recombinant AFP inhibits the phagocytic ability of macrophages. Specifically, AFP induces macrophage polarization towards the M2 phenotype, a state characterized by a reduced capacity for phagocytosis, which allows liver cancer cells to evade the immune response. In addition, the interaction between AFP and macrophages occurs through specific receptors on macrophages, leading to changes in signaling pathways. One critical pathway involved is the PI3K/Akt signaling pathway, which is activated by AFP and further influences macrophage behavior [61,62]. Another study demonstrated that AFP interacts with HuR to enhance the translocation of CD47 to the cell membrane, upregulating the localization of CD47 on the surface of HCC cells and consequently inhibiting macrophages from phagocytizing these cancer cells [63]. In summary, these findings indicate that AFP significantly affects macrophage behavior, promoting a pro-tumorigenic environment by inhibiting their phagocytic activity against hepatoma cells. These insights provide a basis for future therapeutic strategies aimed at improving liver cancer treatment by modulating macrophage function via AFP.

#### 2.5.3. Effects on DCs

AFP can inhibit the maturation of DCs, which are crucial antigen-presenting cells that activate T cells and initiate immune responses. By negatively affecting DC maturation, AFP diminishes the ability of these cells to effectively present antigens, thereby impairing the overall immune response against tumors. Elevated levels of AFP correlate with increased numbers of immature and regulatory DCs, which produce immunosuppressive factors, such as IL-10. This immunosuppressive milieu hinders the maturation and function of DCs, leading to reduced activation of cytotoxic T lymphocytes and a weakened antitumor response [17]. AFP downregulates the expression of costimulatory molecules on DCs, such as CD80 and CD86, which are essential for T cell activation [64]. In summary, AFP plays a crucial role in modulating the functions of DCs, leading to impaired immune activation and contributing to the immune evasion observed in HCC. Consequently, understanding these interactions is critical for developing potential immunotherapeutic strategies targeting the AFP-DC axis, which could enhance the effectiveness of HCC treatments.

#### 2.5.4. Effects on Natural Killer (NK) Cells

Different forms of AFP may exert distinct influences on NK cells. While native alpha-fetoprotein (nAFP) inhibits the cytolytic function of NK cells, tAFP induces their apoptosis. For example, mouse NK cells pretreated with nAFP demonstrate a reduced ability to kill target cells, whereas tAFP induces apoptosis in NK cells in vitro through binding to various hydrophobic low-molecular-weight (LMW) ligands, including metabolites and other small molecules [65]. Additionally, nAFP indirectly inhibits NK cells by regulating the function of DCs and reducing the production of key cytokines such as IL-12, which hinders NK cell activation. To sum up, AFP exerts both direct and indirect effects on NK cells, and these interactions are relevant to immune evasion in cancer [66]. Therefore, understanding these mechanisms may provide valuable insights into potential immunotherapeutic strategies for HCC. The effects of AFP on various immune cells and the associated underlying mechanisms are listed in Table 1.

### 2.6. Resistance to Therapy

Cancer therapy resistance occurs when cancer cells become resistant to treatment. This is a significant cause of treatment failure and cancer recurrence [67,68,69]. AFP contributes to the development of resistance to anticancer therapies through various mechanisms. It can influence the expression and activity of drug transporters on the cell membrane, such as P-glycoprotein. Increased expression of these transporters leads to an enhanced efflux of chemotherapeutic agents from cells, thereby reducing their intracellular concentrations and efficacy [70]. Moreover, AFP promotes cell survival pathways and inhibits programmed cell death by regulating the levels of intrinsic apoptosis markers, such as Bax and Bcl2 [27]. This helps cancer cells resist the cytotoxic effects of chemotherapy. Additionally, AFP may alter the tumor microenvironment by promoting chronic inflammation and recruitment of immunosuppressive cells, creating a more favorable environment for tumor cells to evade the effects of therapies [1]. Furthermore, it can induce metabolic adaptations, such as enhanced glycolysis and altered energy metabolism, in cancer cells. These changes provide energy and intermediates necessary for survival during treatment [47]. Finally, AFP promotes EMT, a process that increases the invasive and migratory potential of cancer cells. EMT is associated with the acquisition of stem cell-like properties that can further enhance resistance to chemotherapy and lead to tumor recurrence. Therefore, through these diverse mechanisms, AFP plays a significant role in promoting resistance to anticancer therapies, complicating treatment strategies, and potentially leading to poor clinical outcomes. Figure 2 summarizes the functions of AFP in relation to HCC progression.

## 3. Role of AFP in Other Cancer Types

AFP levels are closely associated with tumor size, histological differentiation, and vascular invasion in HCC, making them useful for staging the disease. Higher AFP levels are generally linked to greater tumor aggressiveness and poorer prognosis. In particular, AFP serves as a valuable marker for assessing disease progression and provides critical information for establishing treatment strategies in HCC [71,72]. AFP levels are also important in selecting candidates for liver transplantation. Elevated AFP levels have been reported to increase the risk of tumor recurrence after transplantation, and various criteria have been proposed to evaluate transplant eligibility based on AFP levels [73]. Additionally, AFP is a useful biomarker for evaluating treatment response in HCC. An increase or sustained elevation of AFP levels after treatment may indicate treatment failure or tumor recurrence [74].

Importantly, the role of AFP in tumor progression, prognostic prediction, and metastatic potential extends beyond HCC and has been observed in various other cancer types. Elevated AFP levels are used for diagnosis, monitoring, and prognostic evaluation in testicular, ovarian, gastric, colorectal, pancreatic, and lung cancers. Therefore, AFP is no longer solely a diagnostic marker for liver cancer but also a crucial element in understanding cancer biology and developing treatment strategies for other cancers expressing this protein.

### 3.1. Testicular Cancer

A large cohort study of approximately 1500 patients with testicular cancer revealed that over 60% of individuals with nonseminomatous germ cell tumors exhibited elevated AFP levels. This elevation may indicate the presence of specific testicular tumor subtypes associated with a higher malignant potential [75]. Additionally, AFP levels can assist in staging testicular cancer, with higher levels often correlating with more advanced disease, thereby influencing treatment planning and surgical approaches. Post-treatment monitoring of AFP levels is also crucial for the early detection of potential recurrence [19]. However, owing to limitations in the sensitivity and specificity of AFP, it cannot replace tissue diagnosis in the management of testicular cancer [76,77,78]. Consequently, further research, potentially involving the integration of other biomarkers, is needed to more accurately assess the impact of elevated serum AFP levels on cancer progression, recurrence potential, and the need for additional therapies in testicular cancer.

### 3.2. Ovarian Cancer

AFP is a valuable marker for ovarian germ cell tumors, and its levels play a significant role in both diagnosis and prognostic evaluation. An increase in AFP levels indicates the presence of malignant germ cell tumors and is useful for monitoring disease progression. Changes in AFP levels following treatment also serve as an important indicator for evaluating the treatment response [79]. One study found that patients with consistently negative AFP levels for more than a year after treatment did not experience recurrence. However, if AFP levels rise again, recurrence should be considered, even in the absence of clinical signs [80]. Therefore, AFP is a crucial marker for the diagnosis and recurrence assessment of malignant ovarian germ cell tumors. In contrast, the most common form of ovarian cancer, epithelial ovarian cancer (EOC), rarely produces AFP compared to ovarian germ cell tumors. However, when it does, it is considered highly malignant and has a poor prognosis, even with early diagnosis. In a case review of 27 EOC patient samples, EOC-producing AFP was identified in three patients, with particularly high AFP levels observed in specific subtypes, such as clear cell and mucinous carcinoma. This suggests unique clinical challenges and potentially different prognostic outcomes compared to those of typical EOC. In many patients with ovarian cancer, elevated AFP is associated with an increase in CA-125, a widely used tumor marker for ovarian cancer, and appears to be more sensitive than CA-125 for tumor tracking, suggesting that further investigation is needed to explore this possibility [20,81]. Therefore, to gain a clearer understanding of the management of AFP-producing EOC and develop effective treatment strategies, it is essential to continuously monitor AFP levels and conduct in-depth research on the function of AFP in ovarian cancer.

### 3.3. Gastric Cancer

Although AFP is not commonly used as a biomarker for gastric cancer, elevated levels have been observed in some cases, particularly in the advanced stages. High AFP levels in gastric cancer often correlate with a poorer prognosis, larger tumor size, and increased risk of metastasis. AFP-producing gastric cancer is considered a highly aggressive subtype, characterized by a high frequency of early liver and lymph node metastases [82,83]. Mechanistically, AFP facilitates the migration and invasion of gastric cancer cells, potentially by upregulating the expression of metastasis-associated in colon cancer 1 (MACC1) [21]. Transcriptome sequencing studies have indicated that genes highly expressed in AFP-producing gastric cancer are associated with the activation of various oncogenic pathways, whereas genes with low expression are often involved in immune responses. Furthermore, single-sample gene set enrichment analysis has revealed that AFP overexpression in AFP-producing gastric cancer significantly suppresses CD8^+^ T cell infiltration into the tumor microenvironment [84]. While the precise role of AFP in gastric cancer is still under investigation, current evidence suggests it influences tumor growth, immune evasion, and metastasis, similar to its known function in HCC.

### 3.4. Colorectal Cancer

AFP production in colorectal cancer is rare; however, AFP-producing colorectal cancer tends to be more aggressive than typical colorectal cancer and is associated with rapid progression of liver metastasis [22,85]. For example, AFP-producing colorectal cancer in a 47-year-old woman metastasized to the liver and systemic lymph nodes, leading to death [86]. Similarly, the same type of cancer in a 40-year-old male quickly metastasized to the stomach and liver, resulting in fatality [87]. This suggests that AFP production may directly correlate with cancer progression and metastasis. Furthermore, an analysis of 20 cases indicated that the clinicopathological features of AFP-producing colorectal cancer differ from those of typical colorectal cancer, with these patients exhibiting a higher metastasis rate and lower survival rate [88]. Consequently, AFP production is an abnormal physiological response in colorectal cancer, which may serve as an important clinical indicator related to the course of the disease and may be useful in diagnostic and therapeutic approaches.

### 3.5. Pancreatic Cancer

AFP-producing pancreatic cancer exhibits unique clinical characteristics, including high malignancy and early metastatic potential, leading to a poor prognosis [89]. Research has shown that AFP-producing pancreatic cancer cells possess self-renewal and differentiation capabilities, and their enhanced tumorigenic potential has been demonstrated through xenotransplantation experiments. Furthermore, it has been confirmed that these cells tend to exhibit resistance to conventional chemotherapy agents. The elucidated molecular mechanisms reveal that the expression of ATP-binding cassette subfamily A member 12 (ABCA12) transporter in AFP-producing cells is more than twice that of non-AFP-producing cells. ABCA12 is associated with cellular drug metabolism and the efflux of substances, which may explain why these cells exhibit resistance to chemotherapeutic agents [90]. This emphasizes the significant challenges associated with treating this subtype of pancreatic cancer. Therefore, while AFP production is a marker of aggressive disease, further investigation into its role and the associated resistance mechanisms may identify novel therapeutic targets to improve treatment responses in patients with AFP-producing pancreatic cancer.

### 3.6. Lung Cancer

Overexpression of AFP has been identified in certain rare subtypes of lung cancer, suggesting its potential utility in both diagnosis and prognosis. Hepatoid adenocarcinoma of the lung (HAL) is an extremely rare subtype of lung cancer that originates in the lung but exhibits histological and molecular characteristics similar to HCC. In a reported case, the patient’s serum AFP level was extremely elevated, reaching several thousand ng/mL. Histologically, the tumor showed hepatoid differentiation with positive immunohistochemical staining for AFP, HepPar-1, and GPC3, indicating a tumor profile highly reminiscent of liver cancer. Thus, AFP may serve as a useful diagnostic biomarker for HAL, which is typically associated with a poor prognosis [91]. Another study demonstrated the predictive value of AFP in assessing ocular metastasis in lung cancer. The results showed that patients with serum AFP levels above 0.54 ng/mL had a significantly increased risk of ocular metastasis, and AFP exhibited high sensitivity (98.3%), suggesting excellent detection capability [92]. In addition, a study involving non-small cell lung cancer patients showed that decreased expression of miR-668-3p was associated with elevated levels of AFP and other tumor markers. Notably, AFP levels were significantly correlated with clinical indicators of cancer progression, including advanced tumor stage and lymph node metastasis [93]. In conclusion, although AFP is not commonly used as a biomarker in routine lung cancer diagnosis, it may act as a key diagnostic indicator in certain subtypes and play a supportive role in predicting metastasis and assessing tumor progression. Especially when used in combination with other tumor markers or molecular indicators, AFP holds potential for improving the accuracy of both diagnosis and prognosis in lung cancer. The role of AFP in various cancers is summarized in Figure 3.

## 4. The Prospects of AFP-Targeted Anticancer Therapies

Current treatment options for cancer, such as surgery, chemotherapy, and radiation therapy, have limited efficacy and often result in serious side effects [94,95,96]. Therefore, there is an urgent need for more effective targeted therapies, necessitating diverse and in-depth studies for the development of AFP inhibitors. AFP-targeted therapies hold promise for selectively targeting cancer cells expressing high levels of AFP, minimizing damage to healthy cells, reducing side effects, and potentially maximizing the effectiveness of immunotherapy.

### 4.1. Current Status of AFP-Targeted Therapies

#### 4.1.1. AFP-Inhibiting Fragments (AIFs)

AIFs inhibit AFP’s physiological or pathological actions by binding to it and blocking its interaction with receptors. Consequently, AIFs are being thoroughly studied to selectively target tumor cells for cancer treatment, maximize therapeutic effects, and minimize side effects [97]. Although AIFs are currently rarely used as commercial drugs in clinical trials, various AFP-related therapies are actively being explored at the research and development levels. AIFs consist of AFP-derived growth inhibitory peptides (GIPs) and their analogs that, while not binding to the AFP receptor, can enter and affect the enzymatic activity of tumor cells. Newly developed peptide AIFs can also function as antagonists to prevent malignancy mediated by the binding of AFP to its receptor (AFPR) or signaling molecules [98,99]. AIFs include not only peptides but also protein fragments derived from domain-3 of AFP, such as AFP-3BC, rAFP3D, and r3dAFP. These fragments can be endocytosed by cancer cells with high AFPR expression to deliver drugs, thereby blocking the activation of cellular pathways responsible for the growth of cancer cells [100]. Additionally, by targeting myeloid-derived suppressor cells (MDSCs) with full-length AFP or its fragments, it is possible to improve the immunosuppressive environment and promote the activation of NK cells and other immune cells. MDSCs are immunosuppressive cells abundant in cancer patients that contribute to cancer development and progression by inhibiting immune responses. AFP can bind to AFPR-positive MDSCs, and some studies have shown that conjugating AFP to specific toxins can kill MDSCs, resulting in the release of NK cells and cytotoxic lymphocytes, thereby enhancing immune responses against cancer cells [100,101].

#### 4.1.2. AFP-Chimeric Antigen Receptor (CAR) T Cells 

CAR T-cell therapy is an immunotherapy that effectively targets cancer cells containing specific antigens [102,103]. Research is ongoing to target AFP in patients with liver cancer. In one study, CAR was inserted into the T cell gene to recognize the AFP-MHC complex through a specific receptor, allowing T cells to effectively attack AFP-expressing cells. The efficacy of these AFP-CAR T cells was evaluated in animal models, demonstrating their ability to suppress AFP-positive liver cancer cells [104]. Additionally, the expression of immune-related cytokines, such as IFN-γ and IL-2, was measured after the injection of CAR T cells to confirm T cell activation, and it was observed that interactions with tumor cells enhanced the activation of NK cells and other immune cells [104,105]. Thus, to clinically apply AFP-targeted CAR T-cell therapy in the future, its safety and efficacy must be assessed in diverse patient populations. Also, to maximize the effects of this novel treatment, combination trials with other immunotherapies or chemotherapeutic agents should be tried out.

#### 4.1.3. AFP Vaccine

Various types of AFP-based cancer vaccines have been developed to date. Traditionally, these methods have aimed to induce an immune response by either including specific peptides or proteins derived from AFP or by injecting DNA containing the AFP gene to facilitate its synthesis within the body [106,107,108]. Additionally, vaccines can be constructed using specific peptide combinations with immunogenic potential to promote immune cell activation, and there are vaccines that directly incorporate the AFP protein to prompt the immune system to recognize and respond to it [108]. Furthermore, by applying these methods, immunizing Balb/c mice with a recombinant AFP/HSP70 vector resulted in a much stronger T cell response and improved protective effects against AFP-expressing tumors compared to AFP alone [109]. Additionally, combining AFP vaccination with immune checkpoint inhibitors enhanced the immune response, ultimately leading to the suppression of liver cancer growth in mice [110]. These preclinical results highlight the potential of AFP-based vaccines for immunotherapy and provide important foundational data for inducing immune responses in the treatment of HCC and other cancers. However, while AFP vaccination may contribute to extended survival in patients with advanced cancer, full-length glycosylated AFP has been reported to exert immunosuppressive effects that can directly promote cancer growth and MDSCs in some cancers [97,111,112]. Therefore, its use in humans is currently prohibited, and it is essential to consider these limitations in the future development of AFP vaccines for human applications to ensure safety. Figure 4 illustrates the various types of AFP-targeted therapies mentioned above.

### 4.2. Future Perspectives on AFP-Targeted Anticancer Therapies

The development of AFP-targeting therapeutics poses several challenges. Although AFP is highly expressed in HCC and other tumors, it is also expressed at low levels in normal cells [113]. This raises the possibility that targeted therapies can affect normal cells and cause adverse effects. Additionally, AFP-expressing tumors exhibit high heterogeneity, indicating that AFP expression and related pathways can vary among patients, making it difficult to develop universally effective treatments [114,115]. Nonetheless, given that AFP is known to play a role in liver cancer, germ cell tumors, and other types of tumors, AFP-targeted cancer therapies are emerging as a particularly promising field of research. Thus, in addition to current strategies that primarily focus on attacking tumor cells expressing AFP, there is a need to develop new anticancer drugs that directly target AFP or its signaling pathways, such as small molecules and engineered antibodies. Furthermore, given that AFP has been identified as having significant value in tumor immunotherapy, the development of vaccines and CAR T cell therapies targeting AFP should be accelerated. For example, customized vaccines should be developed according to the patient’s immune response and genetic background, and the combination and dose that can achieve the best efficacy should be studied. In addition, their safety and toxicity should be thoroughly evaluated, and methods to minimize side effects should be sought. Finally, it is necessary to combine AFP-targeting therapy with other treatment modalities, such as immunotherapy, chemotherapy, and radiotherapy, to find the optimal treatment combination for treating AFP-expressing tumors. These various attempts to target AFP are expected to improve treatment outcomes, minimize side effects, and improve the quality of life of cancer patients.

## 5. Conclusions

This review demonstrates a direct association between elevated AFP levels and aggressive tumor phenotypes, highlighting its significant predictive value for clinical outcomes. Additionally, we emphasize the importance of AFP as a therapeutic target by specifically discussing its effects on tumor progression, metastasis, immune evasion, and treatment resistance across various cancers, including HCC and testicular, ovarian, gastric, colorectal, and pancreatic cancers. This review also analyzes the current landscape of AFP-targeted therapy research and predicts future development prospects. However, AFP has several limitations as a diagnostic biomarker. AFP levels can be elevated not only in HCC but also in various non-malignant conditions such as chronic hepatitis B and C, liver cirrhosis, acute hepatitis, and pregnancy. This makes it difficult to distinguish HCC from benign liver diseases, thereby increasing the risk of misdiagnosis. In addition, discrepancies between AFP levels and tumor size or stage may occur, requiring cautious interpretation when using AFP for staging or prognostic evaluation [9,116,117]. Despite these limitations, current AFP-targeted therapies remain of interest due to AFP’s critical role in cancer progression. Consequently, ongoing research into the underlying mechanisms of AFP is essential for improving patient outcomes and developing effective treatment strategies. In particular, optimizing the specificity and efficacy of AFP-targeted therapies will present significant opportunities for innovative changes in treating HCC and other AFP-related malignancies.

## Figures and Tables

**Figure 1 ijms-26-04863-f001:**
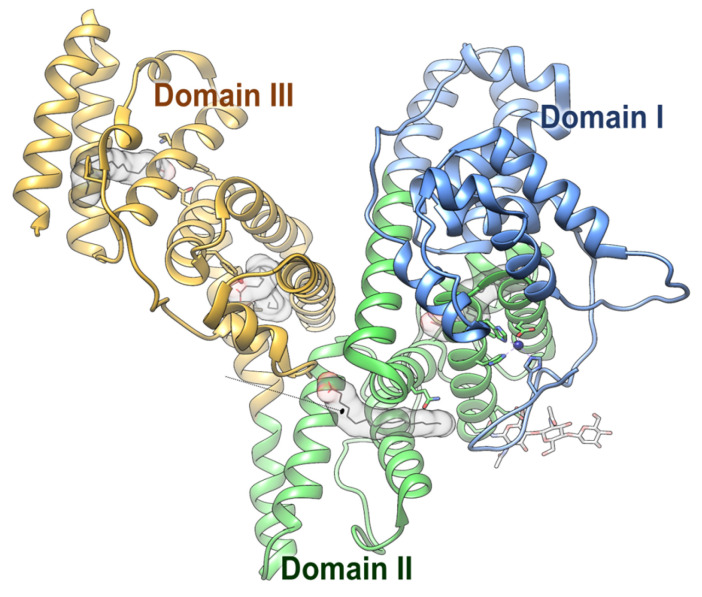
AFP exhibits a V-shaped configuration consisting of three distinct domains. Domain I (blue, N-terminal) is positioned on the right arm of the V, Domain III (yellow, C-terminal) on the left, and Domain II (green, central region) forms the base. The structure was generated based on the Cryo-EM structure with PDB ID 8X1N.

**Figure 2 ijms-26-04863-f002:**
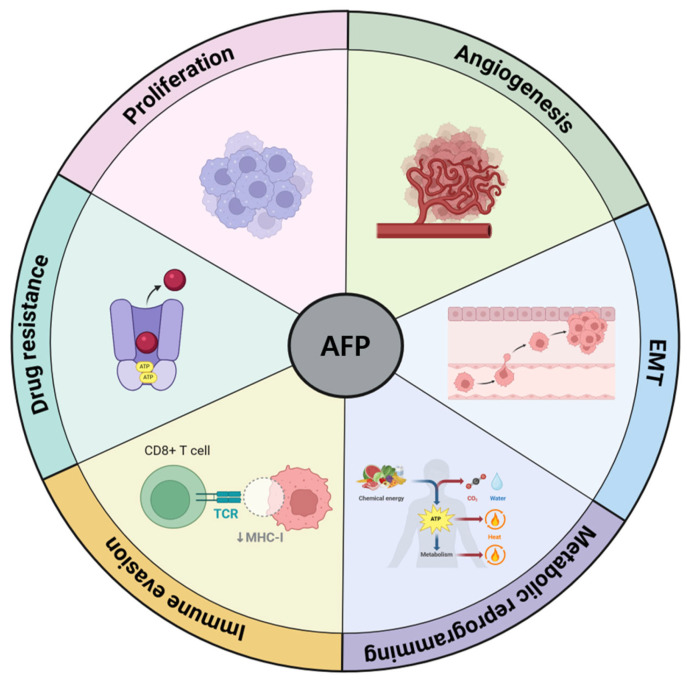
AFP promotes the progression of HCC through several mechanisms, including cell proliferation, angiogenesis, EMT, metabolic reprogramming, immune evasion, and the development of drug resistance.

**Figure 3 ijms-26-04863-f003:**
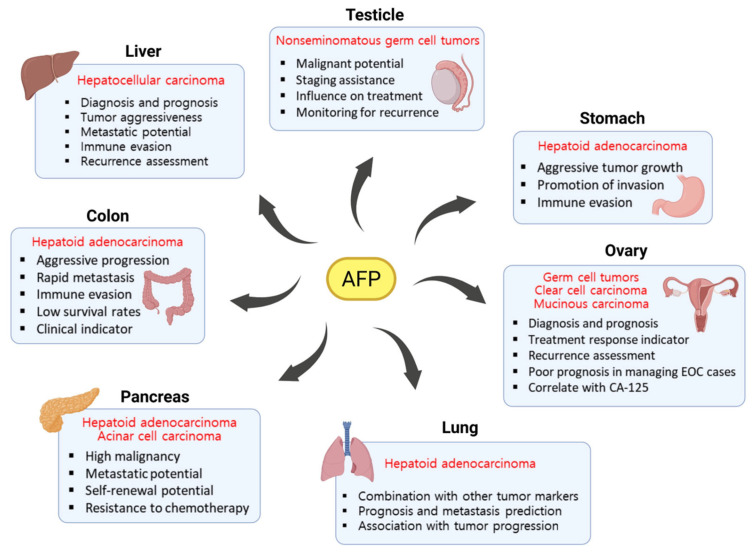
AFP plays an important role in cancers arising in organs beyond the liver.

**Figure 4 ijms-26-04863-f004:**
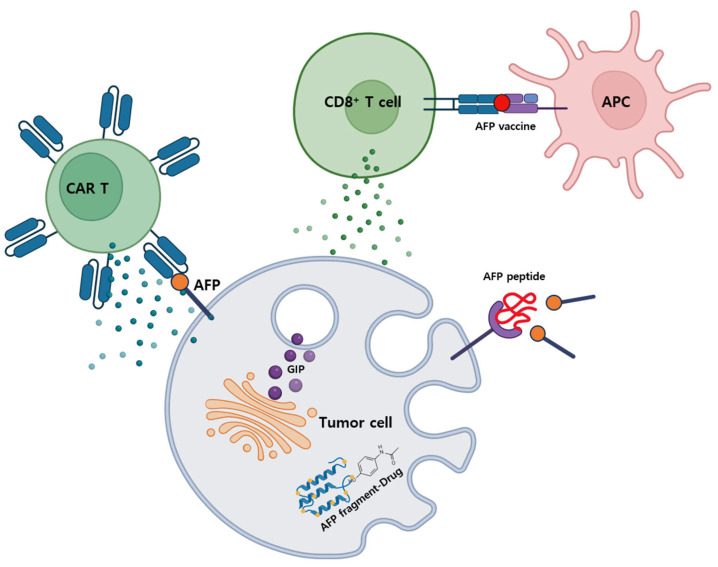
AFP-targeted therapies for cancer treatment include methods such as AIFs, CAR T, and vaccine.

**Table 1 ijms-26-04863-t001:** The effects of AFP on immune cells.

Cell Type	Mechanism	Effect	Outcome
T cells	- Development of CD4^+^ T cells into TGF-β-producing cells	- Enhancement of Treg cell differentiation	- Immune-suppressive environment
Macrophages	- Involvement in the PI3K/Akt signaling - Interaction with HuR - CD47 translocation	- Inhibition of the phagocytic ability - Induction of polarization to M2	- Pro-tumorigenic environment
DCs	- Downregulation of co-stimulatory molecule expression (ex.CD80, CD86) - Modulation of IL-12 secretion	- Inhibition of DCs maturation - Reduction in antigen-presenting ability	- Impaired immune activation - Immune evasion
NK cells	- Induction of apoptosis by binding to hydrophobic LMW ligands (direct) - Inhibition of DCs maturation and reduction in IL-12 (indirect)	- Induction of NK cells apoptosis - Inhibition of NK cell cytolytic function	- Immune evasion

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
