# Peer review of "Exploring the Multifunctional Role of Alpha-Fetoprotein in Cancer Progression: Implications for Targeted Therapy in Hepatocellular Carcinoma and Beyond"

_ijms, 2025, doi:10.3390/ijms26104863_

Round 1
Reviewer 1 Report
Comments and Suggestions for Authors
- First, this is a good review.
- Line 27: I suggest talking about the structure of alpha-fetoprotein and half-life.
- Line 34: Could you please mention lens culinaris agglutinin-related AFP-L1/L2/L3, which was suggested for clinical use in Japan?
- Line 203-205: (not in English)
- Line 232: at least 125 papers are talking about “lung/pulmonary” cancer, 28 talking about thyroid cancer and 58 about uterine/uterus with high AFP, are these organs indicated to be mentioned?
- Line 243: AFP levels can assist in staging testicular cancer à no mention of the role of AFP in staging HCC, both in the diagnosis, and treatment (pre-transplant or as an endpoint), and should mention more about the utility of AFP in clinical practice (Galle, P.R., et al., Biology and significance of alpha-fetoprotein in hepatocellular carcinoma. Liver Int, 2019. 39(12): p. 2214-2229)
- Line 321: [this is the most important] Suggest Figure 2. having (1) label the organ in English ; (2) mention the type of cancer in each organ, such as germ cell tumors; hepatoid adenocarcinoma of stomach and pancreas or acinar cell carcinoma of the pancreas; (3) expands the organs to six or more ?, including Liver (HCC) for the whole opinion at a glance to this paper (also for the convenience to reader in preparing a slide)
Author Response
Comments 1: First, this is a good review.
Response 1: We sincerely appreciate the reviewer’s positive evaluation and are grateful for the kind acknowledgment that this is a good review. This encouragement is highly motivating and has strengthened our commitment to further improving the manuscript. In response to the subsequent constructive comments and suggestions, we have carefully addressed each point in detail below and revised the manuscript accordingly.
Comments 2: Line 27: I suggest talking about the structure of alpha-fetoprotein and half-life.
Response 2: Thank you for the helpful comments. As the reviewer suggested, we incorporated the following sentences in the Introduction section “Alpha-fetoprotein (AFP) is structurally a protein consisting of a single polypeptide chain of 609 amino acids with a molecular weight of approximately 69 kDa, and it shares a structural similarity with albumin. This protein is divided into three structural domains: the N-terminal domain (Domain I, amino acids 1–210), the central domain (Domain II, amino acids 211–402), and the C-terminal domain (Domain III, amino acids 403–609). These three domains are connected by disulfide bonds, forming a V-shaped structure, and each domain carries out specific biological functions. In particular, the C-terminal domain contains a leucine zipper-like structure that allows it to bind to signaling proteins and receptors, thereby promoting tumor growth, mediating intracellular transport, and regulating cellular signaling functions. Figure 1 provides a schematic overview of the structure of human AFP.” (line 27-37, page 1) and “Its serum level generally shows a half-life of approximately five days.” (line 45-46, page 2).
Comments 3: Line 34: Could you please mention lens culinaris agglutinin-related AFP-L1/L2/L3, which was suggested for clinical use in Japan?
Response 3: Thank you for the valuable comment. We also incorporated the following sentences in the Introduction section “AFP is a glycoprotein that can be classified into three glycoforms (AFP-L1, AFP-L2, and AFP-L3) based on its binding affinity to lens culinaris agglutinin (LCA). Among these, AFP-L3, which shows strong binding to LCA, is known to be specifically elevated in HCC. In Japan, AFP-L3 has been proposed and is currently used in clinical practice for the diagnosis and prognosis of HCC.” (line 46-50, page 2). Thank you.
Comments 4: Line 203-205: (not in English)
Response 4: Thank the careful and thoughtful comments. The unintended text has been removed as suggested.
Comments 5: Line 232: at least 125 papers are talking about “lung/pulmonary” cancer, 28 talking about thyroid cancer and 58 about uterine/uterus with high AFP, are these organs indicated to be mentioned?
Response 5: We sincerely thank the reviewer for the insightful suggestion regarding the potential inclusion of lung, thyroid, and uterine cancers based on the number of publications reporting elevated AFP levels. In response, we thoroughly reviewed the literature to evaluate the relationship between AFP expression and each of these cancer types.
- Lung cancer
AFP-producing lung cancers are indeed rare, but several case reports have been published. In particular, hepatoid adenocarcinoma of the lung (HAL) has been shown to share morphological features with hepatocellular carcinoma and is known to produce AFP in most cases. These tumors typically occur in male patients and are often associated with poor prognosis. Although AFP is not generally used as a diagnostic marker for common types of lung cancer, in rare subtypes such as HAL or other AFP-producing variants, serum AFP levels may provide useful information for diagnosis and treatment monitoring. Based on this literature evidence, we have added a brief discussion of AFP’s potential role in rare lung cancer subtypes in the revised manuscript (line 349-371, page 9-10).
- Thyroid and uterine cancers
Although elevated AFP levels have been reported in thyroid and uterine cancers, the overall association between AFP expression and these tumor types remains extremely limited and inconsistent.
In the case of thyroid cancer, AFP elevation has only been documented in exceedingly rare subtypes, such as hepatoid variants of papillary or follicular thyroid carcinoma. These cases are largely anecdotal and do not reflect the clinical characteristics of thyroid cancer as a whole. Furthermore, current clinical guidelines do not recognize AFP as a diagnostic or prognostic marker for thyroid malignancies. For instance, both the 2015 and 2024 guidelines from the American Thyroid Association (ATA) make no reference to AFP, instead emphasizing the use of thyroglobulin and calcitonin. Likewise, other authoritative bodies such as the European Society for Medical Oncology (ESMO) and the National Comprehensive Cancer Network (NCCN) do not recommend the use of AFP in the clinical management of thyroid cancer. (1. 2024 American Thyroid Association Management Guidelines for Adult Patients with Differentiated Thyroid Cancer. Thyroid. 2024. 2. Filetti, S., et al. Thyroid cancer: ESMO Clinical Practice Guidelines for diagnosis, treatment and follow-up. Ann Oncol. 2019;30(12):1856–1883.)
As for uterine cancer, elevated AFP levels have primarily been observed in extremely rare cases such as hepatoid adenocarcinomas or tumors with yolk sac differentiation. These subtypes represent only a very small fraction of all uterine cancers. The majority of patients with endometrial cancer exhibit serum AFP levels within the normal range, and there is insufficient evidence to support AFP as a clinically meaningful biomarker in these cases. (1. Ishiguro, T., et al., Serum alpha fetoprotein in gynaecologic related malignancies. Zentralbl Gynakol 1980;102(21):1209-12. 2. Otani, T., et al., α-Fetoprotein-producing endometrial carcinoma is associated with fetal gut-like and/or hepatoid morphology, lymphovascular infiltration, TP53 Abnormalities, and poor prognosis: Five cases and literature review. Front Med (Lausanne) 2021:8:799163.)
Therefore, although reports of AFP elevation in thyroid and uterine cancers do exist, the extreme rarity and pathological heterogeneity of these cases render it scientifically inappropriate to generalize or emphasize them alongside malignancies with well-established AFP associations, such as hepatocellular carcinoma and germ cell tumors. For this reason, we have decided not to include thyroid and uterine cancers in the main discussion of the manuscript.
We hope this explanation fully addresses the reviewer’s concerns, and we are grateful for the opportunity to clarify this point.
Comments 6: Line 243: AFP levels can assist in staging testicular cancer à no mention of the role of AFP in staging HCC, both in the diagnosis, and treatment (pre-transplant or as an endpoint), and should mention more about the utility of AFP in clinical practice (Galle, P.R., et al., Biology and significance of alpha-fetoprotein in hepatocellular carcinoma. Liver Int, 2019. 39(12): p. 2214-2229)
Response 6: We agreed with the reviewer’s comment. In accordance with the reviewer’s suggestion, we have referred to the recommended literature and inserted the following sentence into the manuscript. “AFP levels are closely associated with tumor size, histological differentiation, and vascular invasion in HCC, making them useful for staging the disease. Higher AFP levels are generally linked to greater tumor aggressiveness and poorer prognosis. In particular, AFP serves as a valuable marker for assessing disease progression and provides critical information for establishing treatment strategies in HCC. AFP levels are also important in selecting candidates for liver transplantation. Elevated AFP levels have been reported to increase the risk of tumor recurrence after transplantation, and various criteria have been proposed to evaluate transplant eligibility based on AFP levels. Additionally, AFP is a useful biomarker for evaluating treatment response in HCC. An increase or sustained elevation of AFP levels after treatment may indicate treatment failure or tumor recurrence.” (lines 249-259, page 7). Thank you again for the reviewer’s insightful comments.
Comments 7: Line 321: [this is the most important] Suggest Figure 2. having (1) label the organ in English; (2) mention the type of cancer in each organ, such as germ cell tumors; hepatoid adenocarcinoma of stomach and pancreas or acinar cell carcinoma of the pancreas; (3) expands the organs to six or more?, including Liver (HCC) for the whole opinion at a glance to this paper (also for the convenience to reader in preparing a slide)
Response 7: (1) Thank you very much for your pointy comment. As you mentioned, we have presented the names of all organs in English. (2) Thank you for your important comment. We have specified the detailed types of cancers associated with each organ. The revised descriptions are expected to provide clearer and more specific information. (3) Thank the reviewer for the valuable suggestion. In response, we have expanded the list of organs to a total of seven by adding the liver and lung.
Reviewer 2 Report
Comments and Suggestions for Authors
The authors have effectively highlighted the dual nature of alpha-fetoprotein (AFP) as both a biomarker and a potential contributor to hepatocellular carcinoma (HCC) progression. They have clearly explained AFP’s pathological roles such as promoting tumor proliferation, invasion, and immune evasion alongside its utility in diagnostics and therapy. The manuscript is logically organized, and the discussion of potential therapeutic strategies, including AFP-targeted drugs and vaccines supported by recent literature, enhances the depth and translational relevance of the article.
Comments:
1. The authors could enhance the manuscript by including details about the basic structure and physiological functions of AFP, possibly supported by a schematic representation for better clarity.
2. While the authors discuss AFP's diagnostic utility, elaborating on its limitations such as false positives or overlap with benign liver conditions would provide a more balanced and comprehensive perspective.
Author Response
Comments 1: The authors could enhance the manuscript by including details about the basic structure and physiological functions of AFP, possibly supported by a schematic representation for better clarity.
Response 1: Thank you for your valuable comment. As you mentioned, we incorporated the following sentences and schematic overview of the structure of AFP in the Introduction section “Alpha-fetoprotein (AFP) is structurally a protein consisting of a single polypeptide chain of 609 amino acids with a molecular weight of approximately 69 kDa, and it shares a structural similarity with albumin. This protein is divided into three structural domains: the N-terminal domain (Domain I, amino acids 1–210), the central domain (Domain II, amino acids 211–402), and the C-terminal domain (Domain III, amino acids 403–609). These three domains are connected by disulfide bonds, forming a V-shaped structure, and each domain carries out specific biological functions. In particular, the C-terminal domain contains a leucine zipper-like structure that allows it to bind to signaling proteins and receptors, thereby promoting tumor growth, mediating intracellular transport, and regulating cellular signaling functions. Figure 1 provides a schematic overview of the structure of human AFP.” (line 27-37, page 1) and (figure 1, page 2).
Comments 2: While the authors discuss AFP's diagnostic utility, elaborating on its limitations such as false positives or overlap with benign liver conditions would provide a more balanced and comprehensive perspective.
Response 2: Thank you for the insightful comment. In response, we have incorporated a detailed discussion on the limitations of AFP as a diagnostic marker in the Conclusion section of the manuscript. “However, AFP has several limitations as a diagnostic biomarker. AFP levels can be elevated not only in HCC but also in various non-malignant conditions such as chronic hepatitis B and C, liver cirrhosis, acute hepatitis, and pregnancy. This makes it difficult to distinguish HCC from benign liver diseases, thereby increasing the risk of misdiagnosis. In addition, discrepancies between AFP levels and tumor size or stage may occur, requiring cautious interpretation when using AFP for staging or prognostic evaluation.” (line 472-478, page 13).